# Stress, Anxiety, and Depression Levels among University Students: Three Years from the Beginning of the Pandemic

**Dimitrios Kavvadas** [1,2,*] **, Asimoula Kavvada** [1,3] **, Sofia Karachrysafi** [1,2] **, Vasileios Papaliagkas** [3] **, Maria Chatzidimitriou** [3] **and Theodora Papamitsou** [1,2]

1    Post-Graduate Program "Health and Environmental Factors", School of Medicine, Faculty of Health, Aristotle University of Thessaloniki, 541 24 Thessaloniki, Greece; akavvada@auth.gr (A.K.); paptheo@med.auth.gr (T.P.)
2    Laboratory of Histology and Embryology, School of Medicine, Faculty of Health, Aristotle University of Thessaloniki, 541 24 Thessaloniki, Greece
3    Department of Biomedical Sciences, School of Health Sciences, International Hellenic University, 574 00 Thessaloniki, Greece
*    Correspondence: kavvadas@auth.gr

**Abstract:** Background: Three years after the outbreak of the COVID-19 pandemic, psychological distress among college students remains increased. This study assesses stress, anxiety, and depression levels among students of the Aristotle University of Thessaloniki by the end of the third year of the pandemic (November 2022), revealing demographic characteristics and probable stressors. Methods: A questionnaire was distributed in November 2022 via the academic students' e-mails. The evaluation was performed with the DASS21 survey tool. The correlation analysis and the effect size calculation were performed with the *t*-test. Results: The majority of participants were undergraduates, on their first or second academic year, female students (67%), age of 18 to 21, unmarried or single (91%), and vaccinated against COVID-19 infection (83.4%). Severely increased levels of stress, anxiety, and depression (21.3%, 23.3%, and 25.1%, respectively) were measured. The normal and mild levels of stress, anxiety, and depression were 64.0%, 66.5%, and 57.2%, respectively. Female and younger students were at a higher risk of extremely severe stress, anxiety and depression prevalence (ORs up to 2.07, *p*-Values < 0.00001). Participants who were receiving psychological or psychiatric treatment exhibited severe stress, anxiety, and depression levels (ORs above 2.9, *p*-Values < 0.00001). Conclusions: Despite the undeniable withdrawal of the COVID-19 pandemic, the community of the Aristotle University of Thessaloniki presents high stress, anxiety, and depression levels, similar to those reported during the first year of the pandemic (November 2020). Stressors and risk factors were according to the reported literature and previous studies on Greek students. Academic psychological support offices should consider the students' "profile" in order to evaluate properly the potential risk for emotional and psychological distress. Evidence suggest that new technology (virtual reality, tele-psychiatry or tele-support apps and sessions) should also be implemented in universities.

**Keywords:** emotional distress; students; DASS21; COVID-19; gender; Greece

## 1. Introduction

The COVID-19 pandemic began as an outbreak of unknown etiology and managed to dominate the world. The appearance of a mutation in the Spike protein of the virus had tremendously increased its transmissibility during the first outbreak, which led to the rapid worldwide spreading [1].

The unexpectedly high morbidity and mortality, along with the unprecedented health and economic emergency, has led countries to take drastic preventive measures [1,2]. The impact of these measures on overall mental health was immense [3–5]. Specific stressors were reported from the very first months of the pandemic; fear of the prolonged outbreak, fear of infection, and fear of exposure to COVID-19 were the main ones [3]. The reports on

mental and emotional distress were alarmingly high during the first and second year of the pandemic, especially in European universities [4,5]. China reported a probability of 11.0%, 34.9%, and 21.1% for developing acute anxiety, stress, and depression, respectively. In the USA, rising levels of stress and anxiety were also reported among college students during the first year of the pandemic [6–8]. Fear of stigma was a significant stressor, derived not only from the COVID-19 infection, but also from the emotional distress that college students tried to suppress [6]. Comparative studies among the first and second year in Greece recorded a significant increase of anxiety, stress, and depression in university students [4].

There is a global agreement that the female gender should be considered as one of the main risk factors, as it has been corelated with increased stress, anxiety, and depression levels during and before the pandemic [4,8,9]. Reports from universities in the USA and France revealed that females were more likely to display distress and negative emotions during the COVID-19 period [10,11]. A qualitive, comparative study in Italy and the UK revealed additional risk factors that should be considered [12]. More specifically, it was observed that uncertainty about the future is a major stressor in young adults, alongside financial hardship and educational instability [12]. Additionally, students who needed psychological support before and during the pandemic were at higher risk [6]. These observations were similar to Greek universities [13]. Furthermore, it was observed that in Greece, the prevalence of negative emotions and psychological distress were significantly higher compared to Chinese students, but similar to those of students from Mediterranean countries such as Italy and Spain [13].

Chandu et al. reported that early stressors have been effectively addressed by mental health tele-services worldwide [3]. Many studies also advocate the fact that that timely crisis-oriented psychological services are important [6–8], as low perceived social support was significantly associated with increased risks for anxiety and depressive symptoms [7]. It is important to note that the transition to distance learning and the implementation of educational innovations had positive effects on students' mental health [6].

Based on the above literature [4–13], it is essential to assess the students' depression, anxiety, and stress levels to plan for necessary support mechanisms, especially during the "recovery" phase [6]. The present study aims at a three-year comparative evaluation of the Aristotle University of Thessaloniki students' psychological distress, by the end of the third year of the pandemic. According to previous results, there was a rapid increase in severe anxiety, stress and depression levels during the first (2020) and the second (2021) years of the pandemic [4]. The current research hypothesis expected no further increase in stress, anxiety, and depression prevalence due to the pandemic's steady recession.

## 2. Materials and Methods

The study was performed with the Depression Anxiety Stress Scale (DASS21), similar to previous launches [4], and the aim was to assess the probable improvement of stress, anxiety, and depression levels of the Aristotle University of Thessaloniki (AUTh) students in comparison to the previous published data.

### 2.1. Study Sample and Comparison Samples

The study sample included Bachelor (BSc and Medical Students), Master (MSc), and PhD students of the university. Based on Aristotle University of Thessaloniki students' population (92,546 active students by November 2022), at a confidence level of 0.95 and a margin of error 0.03 with the largest standard deviation for a proportion at 0.5, the sample size needed is almost 1055 students. There were 2043 participants recorded in the survey. However, 360 of them were excluded due to incomplete participation. Of the rest (1683 participants), 186 were not students (administrative or academic staff) and therefore excluded. The total number of students eligible to participate in the study was 1497 (an acceptable sample size in order to make proper inferences). In our previous study, the research was conducted at two-time intervals: the first period of the survey took place in

November 2020 (first major peak of the pandemic) and included 2322 participants, while the second was in November 2021 (second peak) with 3160 students [4].

### 2.2. Survey Tools

The hosting platform was the LimeSurvey AUTH, which is the official platform for conducting surveys in Aristotle University of Thessaloniki. Both the AUTh bioethics committee (Bioethics Committee No. 1254 date 20 October 2020) as well as the AUTh Data Protection office granted permission.

The questionnaire was available from 10 November 2022 to 25 November 2022. Participants received the survey invitation personally, through their institutional e-mail ("name"@auth.gr). The participants declared whether or not have understood the purposes of the study and accepted participation before completing the survey.

The questionnaire included two main parts, the first with the demographic information and the second with the DASS21 screening tool (Table 1 and Appendix A), similar to previous investigations in the Aristotle University students' community [4].

**Table 1.** The two parts of the questionnaire.

| Part A | Demographic, academic and other personal questions (1 to 15, Appendix A) |
|--------|--------------------------------------------------------------------------|
| Part B | The Likert-4 DASS21 set of questions (16 to 36, Appendix A) |

The Depression-Anxiety-Stress Scale (DASS21) is a widely accepted screening tool that was introduced 27 years ago and classifies the respondent at normal, mild, moderate, severe or extremely severe range of depression, anxiety and stress independently [4]. The 21 questions are divided into three sets of seven questions for depression, anxiety and stress per set. It consists of three self-report scales designed for the screening of depression, anxiety, and stress [4]. Each of the three DASS21 scales contain seven elements, divided into subscales with similar content. The depression scale assesses discomfort, despair, life devaluation, self-devaluation, lack of interest/engagement, and inaction. The anxiety scale assesses autonomic arousal, signs of stress through skeletal muscle movements, stress-induced anxiety, and the subjective experience of anxiety. The stress scale is sensitive to chronic non-specific stimulation and evaluates the difficulty of relaxation, nervous agitation, upset/agitation, etc. The results can be either normal, mild, moderate, severe, or extremely severe. For stress, 0–7 is the normal range, 8–9 is the mild, 10–12 is the moderate, 13–16 is the severe, and above 17 is the extreme severe range. For anxiety, 0–3 is the normal range, 4–5 is the mild range, 6–7 is the moderate, 8–9 is the severe, and above 10 is the extreme severe range. For depression, 0–4 is the normal range, 5–6 is the mild, 7–10 is the moderate, 11–13 is the severe, and above 14 is the extreme severe range. Contemporary tools have been correlated with DASS21 and the findings were the same [4].

### 2.3. Statistical Analysis

The hypothesis was that the differences between depression, anxiety, and stress experienced by normal individuals and clinical populations were gradually different and that the prevenance of our three variables differs in comparison to previous studies [4]. The gradation was delineated based on the continuous distribution of samples with a value range of 0–21, which consist of the sum of the DASS21 scores (Normal, Mild to Sever, Extreme Severe) [4].

The Cronbach's coefficient alpha and McDonald's omega factor were both calculated at 0.951. The results were reported in Mean and Standard Deviation. Apart from the main tables of the results, there are also tables presented in Appendix B and the Supplementary files. Multiple correlation analysis was performed with the *t*-test and Pearson's chi-square test (Appendix B). The effect size calculation was performed with the Cohen's d (small

approximately at 0.2, medium approximately at 0.5, and large > 0.8). The Odds Ratios (ORs) were also calculated (no correlation if OR equals 1, positively correlation if OR > 1 and, negatively correlation if OR < 1). SPSS version 24.0 (IBM, SPSS Inc., Chicago, IL, USA) and Microsoft Excel (2019) version 16.43 were used.

## 3. Results

### 3.1. Demographic and DASS21

The majority of participants were between the age of 18 and 25 (76%), undergraduates (74%), female students (67%), unmarried or single (91%), and vaccinated against COVID-19 (83%) (Table S1). Almost all of the participating students knew someone from their environment with a positive COVID-19 infection with mild to moderate symptoms (Table S2). Previous psychological or psychiatric treatment was reported by 23% of the students (by November, 2022). Current treatment was reported at 11.4% and intake of psychoactive medication at 3.8% (Table S3).

Only a few students were not vaccinated by November 2022, and the majority of them were undergraduates (Table S3). Additionally, vaccination was not corelated to gender (Tables S4 and S5). Gender-based analysis revealed that female students' fear of an impending lockdown was significantly greater in comparison to men (Table S6).

The scores of the DASS21 revealed a prevalence of stress, anxiety and depression in almost 50% of the sample (Table 2), (Figure 1).

**Table 2.** DASS21 scores of the AUTh students on November 2022 (N = 1497 students).

| Scores | Stress | | | Anxiety | | | Depression | | |
|---|---|---|---|---|---|---|---|---|---|
| | M (±STD) | Md | N (%) | M (±STD) | Md | N (%) | M (±STD) | Md | N (%) |
| Normal | 3.58 (±2.36) | 4.0 | 789 (52.7) | 1.22 (±1.11) | 1.0 | 794 (53.0) | 1.68 (±1.47) | 2.0 | 664 (44.4) |
| Mild | 8.47 (±0.50) | 8.0 | 169 (11.3) | 4.45 (±0.50) | 4.0 | 202 (13.5) | 5.47 (±0.50) | 5.0 | 192 (12.8) |
| Moderate | 10.90 (±0.79) | 11.0 | 221 (14.8) | 6.49 (±0.50) | 6.0 | 152 (10.2) | 8.25 (±1.08) | 8.0 | 265 (17.7) |
| Severe | 14.41 (±1.13) | 14.0 | 190 (12.7) | 8.55 (±0.50) | 9.0 | 103 (6.9) | 11.91 (±0.82) | 12.0 | 127 (8.5) |
| Extreme severe | 18.64 (±1.35) | 18.0 | 128 (8.6) | 13.65 (±2.87) | 13.0 | 246 (16.4) | 17.16 (±2.28) | 17.0 | 249 (16.6) |

M: Mean-Average, STD: Standard Deviation, Md: Median, N: Number of Students.

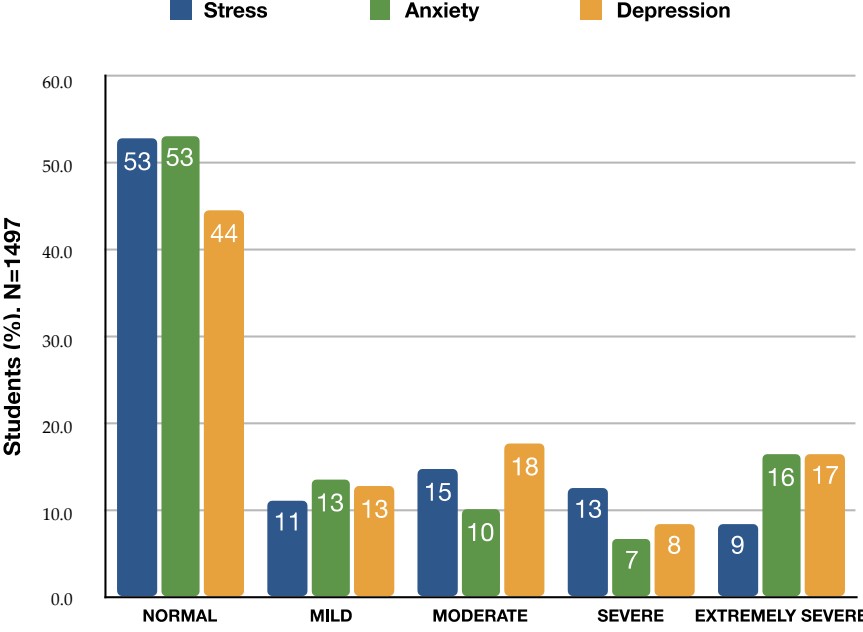

**Figure 1.** Stress, anxiety and depression levels of the AUTh students on November, 2022. Each scale represents the proportion of students for the corresponding level of stress, anxiety and depression.

### 3.2. Multiple Correlation Analysis

The students' demographic characteristics were evaluated in correlation to their psychological distress. Age, gender, marital status, and history of psychological or psychiatric evaluation and/or drug intake were correlated with severe prevalence of stress, anxiety and depression (Appendix B), (Tables 3 and 4).

**Table 3.** The responses of students' demographics in correlation with DASS21 scores (*t*-test).

| Mean (±Standard Deviation) | | Stress | Anxiety | Depression |
|---|---|---|---|---|
| Age range | 18–25 | 8.19 (±5.42) | 5.05 (±4.80) | 7.18 (±5.83) |
| | ≥26 | 6.88 (±5.46) | 3.76 (±4.66) | 5.48 (±5.79) |
| | *t*-test (95% CI) | 1.31 (0.66 to 1.95) ** | 1.30 (0.73 to 1.87) ** | 1.71 (1.02 to 2.40) ** |
| Gender | Male | 6.47 (±5.28) | 3.70 (±4.29) | 5.91 (±5.65) |
| | Female | 8.57 (±5.41) | 5.25 (±4.95) | 7.19 (±5.91) |
| | *t*-test (95% CI) | 2.10 (1.52 to 2.68) ** | 1.55 (1.04 to 2.06) ** | 1.28 (0.65 to 1.91) ** |
| Marital status | Unmarried | 5.90 (±5.15) | 2.80 (±3.93) | 3.96 (±4.71) |
| | Other | 8.07 (±5.45) | 4.93 (±4.84) | 7.05 (±5.89) |
| | *t*-test (95% CI) | 2.17 (1.20 to 3.13) ** | 2.13 (1.28 to 2.97) ** | 3.10 (2.07 to 4.13) ** |
| Cohabitation status | Alone | 8.18 (±5.48) | 4.99 (±4.77) | 7.14 (±6.00) |
| | Not Alone | 7.74 (±5.45) | 4.62 (±4.81) | 6.61 (±5.80) |
| | *t*-test (95% CI) | 0.43 (−0.17 to 1.04) | 0.37 (−0.16 to 0.90) | 0.52 (−0.12 to 1.17) |
| Vaccinated | Yes | 8.12 (±5.49) | 4.89 (±4.86) | 6.96 (±5.95) |
| | No | 6.65 (±5.17) | 3.99 (±4.43) | 5.82 (±5.29) |
| | *t*-test (95% CI) | 1.47 (0.73 to 2.21) * | 0.90 (0.24 to 1.55) * | 1.14 (0.35 to 1.94) * |
| Students | BSc/MD | 7.99 (±5.42) | 4.92 (±4.81) | 6.93 (±5.83) |
| | MSc/PhD | 7.63 (±5.57) | 4.24 (±4.76) | 6.37 (±5.92) |
| | *t*-test (95% CI) | 0.36 (−0.28 to 1.00) | 0.68 (0.12 to 1.24) * | 0.56 (−0.12 to 1.24) |
| Worry/Fear for impending lockdown | Much/Very Much | 9.65 (±6.00) | 6.67 (±5.67) | 8.39 (±6.52) |
| | Not at all/A Little | 7.22 (±5.22) | 4.24 (±4.42) | 6.36 (±5.62) |
| | *t*-test (95% CI) | 2.23 (1.55 to 2.91) ** | 2.43 (1.84 to 3.02) * | 2.03 (1.29 to 2.76) |
| Previous psychological or psychiatric treatment | Yes | 10.21 (±5.67) | 6.73 (±5.32) | 9.22 (±6.34) |
| | No | 7.17 (±5.19) | 4.13 (±4.46) | 6.03 (±5.50) |
| | *t*-test (95% CI) | 3.04 (2.40 to 3.68) ** | 2.60 (2.04 to 3.16) ** | 3.19 (2.50 to 3.87) ** |
| Current psychological or psychiatric treatment | Yes | 11.63 (±5.64) | 7.88 (±5.59) | 10.82 (±6.59) |
| | No | 7.39 (±5.25) | 4.33 (±4.54) | 6.25 (±5.56) |
| | *t*-test (95% CI) | 4.23 (3.39 to 5.08) ** | 3.55 (2.80 to 4.30) ** | 4.57 (3.66 to 5.48) ** |
| Current intake of psychoactive medication | Yes | 13.37 (±5.04) | 9.95 (±5.54) | 13.84 (±5.92) |
| | No | 7.66 (±5.37) | 4.74 (±4.80) | 6.77 (±5.82) |
| | *t*-test (95% CI) | 5.71 (4.29 to 7.13) ** | 5.42 (4.17 to 6.66) ** | 7.35 (5.84 to 8.86) ** |

Statistical significance levels: * $p < 0.05$, ** $p < 0.001$. CI: confidence interval, *t*-test.

Younger students presented extreme severe stress levels at a percentage of 9% and extreme severe anxiety and depression levels at a percentage of 18%. Older students presented lower rates. Female students were significantly more increased in all three variables of stress, anxiety, and depression levels. Cohabitation did not present any correlation with stress, anxiety and depression levels. Stress levels presented a notable augmentation in vaccinated students compared to the unvaccinated ones. Anxiety and depression prevalence was also found in this group of students, but not as increased as stress prevalence. Fear for impending lockdowns was corelated with the severe prevalence of stress, anxiety and depression, especially among female participants (Tables 3 and 4) (Table S6).

**Table 4.** Cohen's d effect size calculation of students' demographics and DASS21 responses (*t*-test, Table 3).

| Predictors | | Stress | Anxiety | Depression |
|---|---|---|---|---|
| Age range | 18–25 / ≥26 | 0.24 | 0.27 | 0.29 |
| Gender | Male / Female | 0.41 | 0.33 | 0.22 |
| Marital status | Unmarried / Other | 0.41 | 0.48 | 0.58 |
| Cohabitation status | Alone / Not Alone | 0.29 | 0.08 | 0.09 |
| Vaccinated | Yes / No | 0.26 | 0.19 | 0.20 |
| Students | BSc/MD / MSc/PhD | 0.07 | 0.14 | 0.10 |
| Worry/Fear for impending lockdown | Much/Very Much / Not at all/A Little | 0.43 | 0.48 | 0.34 |
| Previous psychological or psychiatric treatment | Yes / No | 0.56 | 0.53 | 0.54 |
| Current psychological or psychiatric treatment | Yes / No | 0.78 | 0.70 | 0.75 |
| Current intake of psychoactive medication | Yes / No | 1.09 | 1.01 | 1.20 |

Odds ratios also revealed that younger students are at higher risk of stress, anxiety and depression. Female gender is a risk factor for mild to severe stress and for extremely severe stress, anxiety and depression. Unmarried students (in their majority younger, undergraduates) were at higher risk in comparison to married ones. Students who reported living alone presented a higher risk of extremely severe stress, anxiety and depression. Students who were not vaccinated against COVID-19 infection were not at high risk for stress, anxiety and depression in comparison to those who were vaccinated. Significantly higher levels of stress, anxiety, and depression were observed among students who received psychological or psychiatric treatment (Tables 5 and 6).

**Table 5.** Mild to severe prevalence of stress, anxiety and depression, odds ratios (ORs).

| Mild to Severe Scales | Stress | Anxiety | Depression |
|---|---|---|---|
| Age 18–25 vs. ≥26 | 1.40 | 1.57 | 1.55 |
| Female vs. Male | 1.93 | 1.26 | 1.14 |
| Unmarried vs. Other | 1.62 | 1.46 | 1.46 |
| Vaccinated: Yes vs. No | 1.44 | 1.02 | 1.04 |
| Previous psychological or psychiatric treatment Yes vs. No | 1.75 | 1.36 | 1.19 |
| Current psychological or psychiatric treatment Yes vs. No | 1.82 | 1.48 | 0.94 |
| Current intake of psychoactive medication Yes vs. No | 1.80 | 0.97 | 0.65 |

Regarding the academic characteristics and DASS21 results, undergraduates were more stressed, anxious and depressed in comparison to the postgraduates and PhD candidates. However, the results were not significant.

**Table 6.** The extreme severe prevalence of stress, anxiety and depression odds ratios (ORs).

| Extreme Severe Scale | Stress | Anxiety | Depression |
|---|---|---|---|
| Age: 18–25 vs. ≥26 | 1.33 | 1.60 | 1.44 |
| Female vs. Male | 1.62 | 2.07 | 1.66 |
| Unmarried vs. Other | 2.12 | 3.36 | 3.41 |
| Vaccinated: Yes vs. No | 1.55 | 1.40 | 1.49 |
| Previous psychological or psychiatric treatment Yes vs. No | 2.97 | 2.65 | 2.77 |
| Current psychological or psychiatric treatment Yes vs. No | 4.53 | 3.57 | 4.09 |
| Current intake of psychoactive medication Yes vs. No | 5.09 | 6.79 | 9.11 |

## 4. Discussion

The results of the present study consist of an evaluation of the stress, anxiety and depression levels in the AUTh students' community, during the third year of the pandemic. Stress and depression prevalence was similar to the first-year evaluation [4]. Anxiety levels were increased, but not as much as the second-year evaluation on November 2021 [4] (Figure 2).

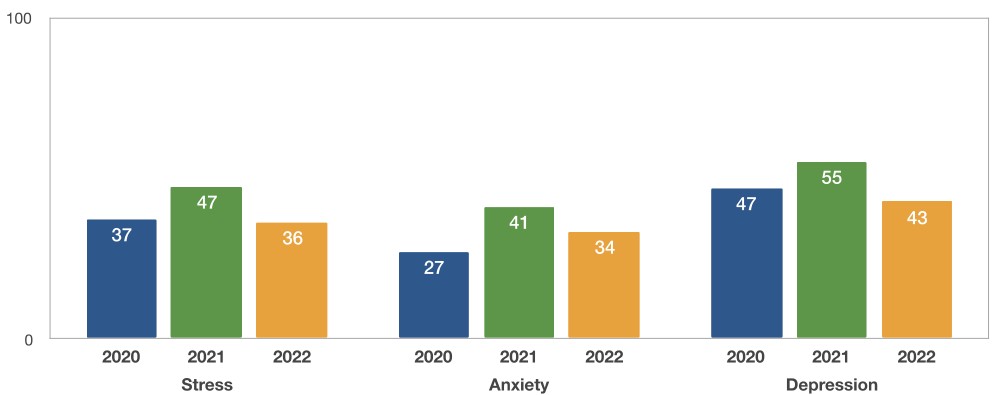

**Figure 2.** Moderate to extremely severe stress, anxiety and depression levels (%) of Aristotle University students from November 2020 to November 2022 [4].

The three-year analysis revealed similar demographic characteristics and correlations. During the third-year analysis, previous psychological or psychiatric treatment was reported as increased during November 2022 in comparison to the first year of the pandemic (November 2020) [4]. An extremely severe prevalence of stress, anxiety and depression was observed during the second year of the pandemic (November 2021), with less than half of the students to be classified in the normal range of stress (40.7%), anxiety (44.4%) and depression (34.6%) [4]. In the present study, the evaluation during November 2022 presented a decline in the extremely severe levels (Table 2). Moderate to extremely severe prevalence among students was chronologically identified as follows:

- November 2020: stress 37.4%, anxiety 27.2%, depression 47.0% [4].
- November 2021: stress 47.3%, anxiety 41.1%, depression 55.0% [4].
- November 2022: stress 36.1%, anxiety 33.5%, depression 42.8% (Table 1).

Students who were receiving psychological or psychiatric treatment exhibited extremely severe stress, anxiety and depression [4]. The negative effects of the pandemic at the in-house-relations were also increased during these two years (2020–2021) [4]. In the current evaluation (November 2022), mild levels of anxiety, stress, and depression were associated with psychiatric treatment. Studies in Greece report similar correlations for these groups before the pandemic [14]. Female participants were more affected in comparison to males (ORs 1.66 to 2.07, *p*-value < 0.00001) (Tables 5 and 6).

It should be noted that the prevalence of stress, anxiety and depression were not substantially increased prior to the pandemic [15–20]. During the years 2009–2011, a mild increase in depression levels was observed in the general population while stress remained stable [17]. There were no specific Greek studies on university students' mental health before the pandemic, apart from a few that focused on possible stressors and habits. For instance, it was observed that Greek students were prone to alcohol consumption which could lead to psychological imbalance and negative feelings [4,14,17].

Recent studies reported comparable findings with the Aristotle University of Thessaloniki. Wong et al. conducted a survey on students (the majority females and undergraduates) and presented a prevalence of moderate to severe depression, anxiety and stress (53.9%, 66.2% and 44.6%, respectively). Exercise was associated with decreased depression levels, and master (MSc) students were associated with reduced stress prevalence [21]. These findings were in complete alignment with ours. Dasor et al. included students across four universities and revealed elevated stress, anxiety and depression levels at percentages of 60.6%, 66.8% and 42.6%, while a meta-analysis of 20 countries resulted in a prevalence of 58%, 50%, and 71%, respectively [22,23].

Aslan et al. conducted a similar large-scale study in Turkey and revealed severe stress, anxiety and depression levels (84%, 36%, and 55%, respectively). The authors focused on similar demographic and psychological characteristics to ours. More specifically, it was reported that religiosity level, marital status, year of study, gender, physical activity, COVID-19 symptoms, death of a close relative and job loss should be considered as major stressors, correlated with the psychological distress of students in Turkey [24]. These interesting observations indicate the importance of analyzing the participants' characteristics to designate probable risk factors.

In Europe, students seemed to present higher stress, anxiety, and depression levels compared to the general population. The reported findings were aligned with ours, as it was also observed that the later phase of the pandemic was associated with lower global stress and anxiety in Europe [25]. Indeed, depression was higher than stress and anxiety in the AUTh students during the third year of the pandemic. Another similar finding was that during the pandemic, females reported higher stress, anxiety and depression levels compared to males. This gender-based effect is already known from the time before the pandemic [26,27]. Therefore, it is to an extent expected from female students to present higher stress, anxiety and depression levels. A Russian cross-sectional study reported a prevalence of depression, anxiety, and stress in more than half of their students [28]. These proportions were quite similar to Greek students. Another cross-sectional study on December 2021 and June 2022 reported moderate anxiety and depression, associated with gender, age, and studies [29].

The vaccination rates of Greek students on November 2022 were similar to those of November 2021 [4]. Regarding vaccination against COVID-19, the majority of students who had not been vaccinated were undergraduates [4]. Students who were vaccinated against COVID-19 infection were at higher risk for stress, anxiety and depression, in comparison to those who did not receive vaccination. Early studies on general populations revealed an elevated willingness to undergo vaccination against the COVID-19 infection [30]. However, healthcare workers presented a decreased interest in receiving the vaccine, in comparison to the general population, and thus, many people became reluctant too [31,32]. Of course, there is a variety of factors that contribute to the acceptance of the vaccination programs (gender, age, education, income, marital status, social media, job, and psychological distress) [33–36]. In Greece, reluctance was exhibited by females and the less-educated respondents [31,32]. Parents exhibited willingness for vaccination, and attitudes of prevention [32]. During the pandemic, a pattern of prevention and protection was observed in several individuals who were vaccinated in the past against the influenza virus [37].

As mentioned, the prevalence of negative emotions and psychological distress were similar to those of students from Mediterranean countries, such as Italy and Spain [13]. These rates were significantly higher compared to Chinese students [13]. Several studies

claim that Eastern Mediterranean Regions (EMR) present higher prevalence of mental disorders and emotional distress on average, even before the pandemic [38–40]. A large systematic review and meta-analysis estimated that the EMR presented the highest pre-pandemic prevalence for depression (14.8%), followed by generalized anxiety disorder (10.4%), post-traumatic stress disorder (7.2%), substance use (4.0%) and obsessive-compulsive disorder (2.8%) [38]. Poor economy and instability, in comparison to the wealthy countries of Europe, seem to be the main reasons [38,40]. The poorest countries in these regions have seen a fall in the available workforce, probably as a consequence of conflict, political instability and displacement [40]. Religion also seems to have an important role, especially regarding the suicidal behavior in the Mediterranean Countries [41]. Therefore, the clinically important anxiety, depressive and post-traumatic stress symptoms that were observed in the Greek general population (part of EMR) during the pandemic were to some extend expected [39]. Additionally, as mentioned before, Greek students are prone to other stressors, such as alcohol [16,17].

Taking this all under consideration, there is indeed a weighted background in Greece that, however, does not justify the extremely high prevalence of stress, anxiety and depression during the pandemic. Interestingly, Dong et al. observed that people who lived in tighter cultural areas presented less psychological disorders due to the pandemic [42]. This moderating effect of cultural tightness was further mediated by perceived protection efficacy during the pandemic [42].

The constant monitoring of the Aristotle University students' community revealed increased levels of stress, anxiety, and depression during the last three years (2020 to 2022) [4]. The 24 h-communication line that was established in the AUTh for members who seek psychological support or counseling is quite essential based on our findings. Furthermore, universities should also consider integrating virtual reality (VR) and digital technologies as coping strategies in order to efficiently support their students [43]. Several studies have already discussed the need for telepsychiatry and VR as a psychological tool for intervention [44–49]. Telepsychiatry is a promising and growing way to deliver mental health services [50]. Digitally delivered cognitive behavioral therapy and mind–body practice techniques have shown to be beneficial strategies against anxiety symptoms [43]. Hatta et al. suggest that VR could provide a three-dimensional (3D) ecosystem for people to participate in interactive environments, assist in the training, evaluation, delivery, and supervision of psychotherapy skills [45]. Especially in quarantine periods, VR could be a rather good substitute for public gyms and private group fitness for physical–psychological wellbeing [51].

*Limitations*

Possible limitations were the lack of a specific independent stressors evaluation and a detailed psychiatric background evaluation of the students, especially regarding the female students who presented higher stress, anxiety and depression levels. In most studies, students are predominantly female, of young age, and possibly possess other risk factors. This could potentially contribute to an explanation regarding the students' high psychological distress in comparison to the general population and must be taken under serious consideration during the interpretation of the results. The authors made an effort to avoid long questionnaires in order to minimize the risk of losing participations and therefore several aspects were not evaluated thoroughly. Additionally, the question about psychological or psychiatric treatment was rather high (23%), perhaps due to the large range of the question. It might be better in future studies to distinguish between "psychological counseling" (i.e., low level treatment) and "therapy" (i.e., more intensive intervention in case of severe impairment).

**5. Conclusions**

To the best of our knowledge, this is the first large three-year evaluation of a specific EMR university during the pandemic. The Aristotle University of Thessaloniki students'

community was greatly affected during the pandemic. The current research revealed no further increase in stress, anxiety, and depression prevalence, perhaps due to the pandemic's steady recession. One should consider that people might have simply adjusted to life during the pandemic. Nevertheless, stress, anxiety, and depression prevalence was reported at similar levels to the first year of the pandemic (November 2020). There is a long way until the prevalence of stress, anxiety, and depression returns to pre-pandemic levels. These findings were in line with international and Greek reports. The factorial analysis of the demographic and social variables indicated the same statistically significant correlations as in the previous two years. The constant screening of psychological distress and implementation of student-centered interventions will help improve the mental well-being of students in order to return to the pre-pandemic state.

**Supplementary Materials:** The following supporting information can be downloaded at: https://www.mdpi.com/article/10.3390/clinpract13030054/s1, Table S1: Demographic characteristics of the students during the 3rd year of completing the questionnaire. Table S2: Questions about COVID-19 during the two years. Table S3: Students' mental health characteristics and social burden due to the pandemic. Table S4: University status of students. Table S5: Vaccination against COVID-19 infection and correlations (November, 2022). Table S6: Concern about impending lockdown in correlation to gender (November, 2022).

**Author Contributions:** Conceptualization, D.K.; methodology, D.K., T.P. and S.K.; validation, A.K., S.K. and V.P.; formal analysis, D.K. and A.K.; investigation, A.K. and D.K.; resources, D.K. and T.P.; data curation, D.K.; writing—original draft preparation, D.K. and A.K.; writing—review and editing, A.K., V.P. and M.C. visualization, D.K. and A.K.; supervision, V.P., M.C. and T.P.; project administration, D.K.; funding acquisition, T.P. All authors have read and agreed to the published version of the manuscript.

**Funding:** This research received no external funding.

**Institutional Review Board Statement:** In accordance with the Declaration of Helsinki, approved by the Bioethics Committee of Aristotle University of Thessaloniki (Bioethics Committee No. 1254 date 20/10/2020).

**Informed Consent Statement:** Informed consent was obtained from all subjects involved in the study.

**Data Availability Statement:** Not applicable.

**Conflicts of Interest:** The authors declare no conflict of interest.

**Appendix A**

**The questionnaire that was distributed to the Aristotle University students.**
*(a): evaluation of anxiety, (s): evaluation of stress, (d): evaluation of depression*
1. Age range
2. Gender
3. Marital Status
4. Health Professional (YES/NO)
5. Cohabitation
6. Changes in professional activity
7. Known person diagnosed positive for COVID-19 (YES/NO)
8. Symptoms manifestation
9. Vaccination against COVID-19 (YES/NO)
10. Concerns about an impending lockdown (0, 1, 2, 3)
11. Psychological or psychiatric treatment in the past (YES/NO)
12. Psychological or psychiatric treatment at this time (YES/NO)
13. Psychotropic drugs intake (YES/NO)
14. Category of students (Undergraduate BSc or MD, MSc, PhD)
15. Year of study (for undergraduate students)
16. I found it hard to wind down (s)
17. I was aware of dryness of my mouth (a)

18. I couldn't seem to experience any positive feeling at all (d)

19. I experienced breathing difficulty (e.g., excessively rapid breathing, breathlessness in the absence of physical exertion) (a)

20. I found it difficult to work up the initiative to do things (d)

21. I tended to over-react to situations (s)

22. I experienced trembling (e.g., in the hands) (a)

23. I felt that I was using a lot of nervous energy (s)

24. I was worried about situations in which I might panic and make a fool of myself (a)

25. I felt that I had nothing to look forward to (d)

26. I found myself getting agitated (s)

27. I found it difficult to relax (s)

28. I felt down-hearted and blue (d)

29. I was intolerant of anything that kept me from getting on with what I was doing (s)

30. I felt I was close to panic (a)

31. I was unable to become enthusiastic about anything (d)

32. I felt I wasn't worth much as a person (d)

33. I felt that I was rather touchy (s)

34. I was aware of the action of my heart in the absence of physical exertion (e.g., sense of heart rate increase, heart missing a beat) (a)

35. I felt scared without any good reason (a)

36. I felt that life was meaningless (d)

## Appendix B

**Table A1.** The responses of students' demographics are in correlation with DASS21 responses (Chi-square analysis, *p*-Value significant at 0.05).

| | | Stress | | | Anxiety | | | Depression | | |
|---|---|---|---|---|---|---|---|---|---|---|
| **Responses of Students** | | **Normal** | **Mild to Severe** | **Extreme Severe** | **Normal** | **Mild to Severe** | **Extreme Severe** | **Normal** | **Mild to Severe** | **Extreme Severe** |
| Age range | 18–25 | 572 | 462 | 103 | 562 | 372 | 203 | 464 | 471 | 202 |
| | ≥26 | 217 | 118 | 25 | 232 | 85 | 43 | 200 | 113 | 47 |
| | *p*-Values | | 0.004 | | | <0.00001 | | | <0.00001 | |
| Gender | Male | 324 | 143 | 31 | 310 | 136 | 52 | 253 | 184 | 61 |
| | Female | 465 | 437 | 97 | 484 | 321 | 194 | 411 | 400 | 188 |
| | *p*-Values | | <0.00001 | | | <0.00001 | | | 0.00027 | |
| Marital status | Unmarried | 699 | 541 | 122 | 699 | 425 | 238 | 579 | 542 | 241 |
| | Other | 90 | 39 | 6 | 95 | 32 | 8 | 85 | 42 | 8 |
| | *p*-Values | | 0.023 | | | 0.00003 | | | <0.00001 | |
| Cohabitation status | I live alone | 228 | 184 | 45 | 232 | 143 | 82 | 194 | 178 | 85 |
| | With 1 person | 205 | 144 | 33 | 206 | 112 | 64 | 176 | 142 | 64 |
| | With 2 or more | 356 | 252 | 50 | 356 | 202 | 100 | 294 | 264 | 100 |
| | *p*-Values | | 0.556 | | | 0.701 | | | 0.547 | |
| Vaccinated | Yes | 634 | 501 | 113 | 652 | 382 | 214 | 541 | 489 | 218 |
| | No | 155 | 79 | 15 | 142 | 75 | 32 | 123 | 95 | 31 |
| | *p*-Values | | 0.0037 | | | 0.197 | | | 0.085 | |
| Previous psychological or psychiatric treatment | Yes | 120 | 171 | 57 | 126 | 124 | 98 | 100 | 147 | 101 |
| | No | 669 | 409 | 71 | 668 | 333 | 148 | 564 | 437 | 148 |
| | *p*-Values | | <0.00001 | | | <0.00001 | | | <0.00001 | |
| Current psychological or psychiatric treatment | Yes | 41 | 88 | 41 | 43 | 65 | 62 | 39 | 64 | 67 |
| | No | 748 | 492 | 87 | 751 | 392 | 184 | 625 | 520 | 182 |
| | *p*-Values | | <0.00001 | | | <0.00001 | | | <0.00001 | |
| Current intake of psychoactive medication | Yes | 10 | 30 | 17 | 9 | 17 | 31 | 5 | 17 | 35 |
| | No | 779 | 550 | 111 | 785 | 440 | 215 | 659 | 567 | 214 |
| | *p*-Values | | <0.00001 | | | <0.00001 | | | <0.00001 | |

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
