# Peer review of "Stress, Anxiety, and Depression Levels among University Students: Three Years from the Beginning of the Pandemic"

_clinpract, doi:10.3390/clinpract13030054_

Round 1

Reviewer 1 Report

Assessing psychological health and risk factors during the COVID-19 pandemic is worthy of scholarly attention. So worthy, in fact, is it, that there are plenty of studies addressing the issue (as you rightly note in the manuscript). It is because there are so many such studies that I flagged the current project as not being terribly new or original. That said, one cannot have too much data, so the results reported here are welcome. I believe the paper could be enhanced if you explicitly gave the data for similar psychological and risk factors prior to the pandemic. You do note that there is a discrepancy, but precise numbers pre- during- and post-pandemic would be helpful. Here are a few extra thoughts that may help make the manuscript stronger…

You note, “it was observed that in Greece, the prevalence of negative emotions and psychological distress were significantly higher compared to Chinese students, but similar to those of students from Mediterranean countries such as Italy and Spain.” This is interesting. Perhaps it goes beyond your expertise, but some speculation as to why this was the case could be helpful, especially when it comes to interventions. For instance, China is a collectivist culture; the Mediterranean cultures tend to be individualist. Might there be a correlation between less severity and collectivism? Does collectivism present greater social support and thus a reduction in risk of psychological disturbances?

You note, “The current research hypothesis expected no further increase in stress, anxiety, and depression prevalence due to the pandemic’s steady recession.” This is certainly a plausible suggestion. Might an alternative be that people simply adjusted to life during the pandemic? In other words, people simply got used to the reality of the pandemic? A normalized threat is a reduced threat?

You note, “Students who were not vaccinated against Covid-19 infection were not at high risk for stress, anxiety and depression in comparison to those who were vaccinated.” Here it seems that we have two groups prior to the pandemic. One group appears rather robust when it comes to the possibility of a public health threat; the other group appears susceptible to the psychological disturbances associated therewith. I think it would be interesting to tap into this distinction, possibly using the 5-factor model of personality? For example, are extraverts less or more likely to get vaccinated? Is there a political dimension to the distinction? Answers to such questions could potentially inform intervention strategies.

On that last note, I found your conclusion a bit underwhelming. You note, “The 24-h-communication line that was established for members who seek psychological support or counseling is quite essential based on our findings. The support centers should also consider integrating virtual reality technologies in order to efficiently support these psychopathological symptoms, especially due to the referring population which is consisted of students, who accept rather easily such innovations.” Aside from the awkward English of the last sentence, the conclusion regarding the necessity of support services doesn’t seem to be anything terribly new. Your suggestion about virtual reality technologies arrives rather abruptly here, and without comment or qualification. Has it been shown in the literature that virtual reality interventions enjoy an increased efficacy? If so, perhaps a word or two on how/why.

Overall, I believe your study contributes to the growing literature on the pandemic and mental health. It basically corroborates what others have found. In this regard, I would like to see some more robust explanations for what we find and more explicit recommendations for how to enhance a global response to mental health issues associated with the public health crisis.

Author Response

Assessing psychological health and risk factors during the COVID-19 pandemic is worthy of scholarly attention. So worthy, in fact, is it, that there are plenty of studies addressing the issue (as you rightly note in the manuscript). It is because there are so many such studies that I flagged the current project as not being terribly new or original. That said, one cannot have too much data, so the results reported here are welcome.

-Thank you for your time and effort to review our manuscript and for your kind suggestions. Indeed, there are plenty of studies on the matter. However, this particular study presents a unique aspect of a three-year comparison of a large university. Thank you for your effort to improve our manuscript! We hope you find our responses satisfactory.  

I believe the paper could be enhanced if you explicitly gave the data for similar psychological and risk factors prior to the pandemic. You do note that there is a discrepancy, but precise numbers pre- during- and post-pandemic would be helpful. Here are a few extra thoughts that may help make the manuscript stronger…

 You note, “it was observed that in Greece, the prevalence of negative emotions and psychological distress were significantly higher compared to Chinese students, but similar to those of students from Mediterranean countries such as Italy and Spain.” This is interesting. Perhaps it goes beyond your expertise, but some speculation as to why this was the case could be helpful, especially when it comes to interventions. For instance, China is a collectivist culture; the Mediterranean cultures tend to be individualist. Might there be a correlation between less severity and collectivism? Does collectivism present greater social support and thus a reduction in risk of psychological disturbances?

-Thank you, we added a mini-discussion paragraph regarding the reasons that Mediterranean countries present higher psychological distress. There are studies which claim that Eastern Mediterranean Regions (EMR) present higher prevalence of mental disorders and emotional distress on average. A large systematic review and meta-analysis estimated that the EMR presented the highest pre-pandemic prevalence for depression (14.8%), followed by generalized anxiety disorder (10.4%), post-traumatic stress disorder (7.2%), substance use (4.0%), obsessive compulsive disorder (2.8%), etc. Poor economy and instability, in comparison to the wealthy countries of Europe seem to be the main reasons. The poorest countries in the region have seen a fall in the available workforce, probably as a consequence of conflict, political instability and displacement. Religion also seems to have an important role, especially regarding the suicidal behavior in the Mediterranean Countries. Therefore, the clinically important anxiety, depressive and post-traumatic stress symptoms that were observed in Greek general population (part of EMR) during the pandemic, were at some extend expected. Also, Greek students are prone to other stressors, such as alcohol. Taking the all the above under consideration, there is indeed a weighted background in Greece which however, does not justifies the extreme high prevalence of stress, anxiety and depression during the pandemic. Interestingly, Dong et al. found that the increase in psychological disorders due to the COVID-19 was less pronounced among people who lived in tighter cultural areas. This moderating effect of cultural tightness was further mediated by perceived protection efficacy during the pandemic.

  • Zuberi A, Waqas A, Naveed S, Hossain MM, Rahman A, Saeed K, Fuhr DC. Prevalence of Mental Disorders in the WHO Eastern Mediterranean Region: A Systematic Review and Meta-Analysis. Front Psychiatry. 2021 Jul 14;12:665019. doi: 10.3389/fpsyt.2021.665019. PMID: 34335323; PMCID: PMC8316754.

  • Karaivazoglou K, Konstantopoulou G, Kalogeropoulou M, Iliou T, Vorvolakos T, Assimakopoulos K, Gourzis P, Alexopoulos P. Psychological distress in the Greek general population during the first COVID-19 lockdown. BJPsych Open. 2021 Feb 24;7(2):e59. doi: 10.1192/bjo.2021.17. PMID: 33622422; PMCID: PMC7925978.
  • Eskin M. Suicidal Behavior in the Mediterranean Countries. Clin Pract Epidemiol Ment Health. 2020 Jul 30;16(Suppl-1):93-100. doi: 10.2174/1745017902016010093. PMID: 33029186; PMCID: PMC7536731.
  • Rahman, A. Mental disorders in the Eastern Mediterranean Region. Int J Public Health 63(Suppl 1), 9–10 (2018). https://doi.org/10.1007/s00038-017-0986-1
  • Dong D, Chen Z, Zong M, Zhang P, Gu W, Feng Y, Qiao Z. What protects us against the COVID-19 threat? Cultural tightness matters. BMC Public Health. 2021 Nov 22;21(1):2139. doi: 10.1186/s12889-021-12161-1. PMID: 34809585; PMCID: PMC8607057.

-More specifically, about the pre-pandemic data in Greece, a mild depression prevalence was revealed due to the beginning of the economic crisis, but the results were in most cases conflicting. Moreover, the fact that students, particularly from Greece, are prone to alcohol consumption during their first years of college, could lead to psychological imbalance and negative feelings. On the contrary, another pre-pandemic study assessed the effects of the Mediterranean diet on Greek university students and found reduced levels of depression and stress, due to the consumption of specific local products. We added the above information on a separate paragraph (Discussion).

  • https://doi.org/10.3390/ijerph120504709.
  • http://www.ncbi.nlm.nih.gov/pubmed/24486974.
  • https://doi.org/10.1080/07448481.2021.1996369.
  • https://doi.org/10.1080/07448481.2018.1432625.
  • https://doi.org/10.1002/hpm.2881.
  • https://doi.org/10.1177/0020764019864122.
  • https://doi.org/10.22365/jpsych.2021.025.

You note, “The current research hypothesis expected no further increase in stress, anxiety, and depression prevalence due to the pandemic’s steady recession.” This is certainly a plausible suggestion. Might an alternative be that people simply adjusted to life during the pandemic? In other words, people simply got used to the reality of the pandemic? A normalized threat is a reduced threat?

-Thank you for your very interesting comments. We added the alternative suggestion regarding the stability of the stress, anxiety and depression levels during the third year of the pandemic in our conclusive paragraph of the discussion section.

You note, “Students who were not vaccinated against Covid-19 infection were not at high risk for stress, anxiety and depression in comparison to those who were vaccinated.” Here it seems that we have two groups prior to the pandemic. One group appears rather robust when it comes to the possibility of a public health threat; the other group appears susceptible to the psychological disturbances associated therewith. I think it would be interesting to tap into this distinction, possibly using the 5-factor model of personality? For example, are extraverts less or more likely to get vaccinated? Is there a political dimension to the distinction? Answers to such questions could potentially inform intervention strategies.

-Thank you! We added the below paragraph in discussion in order to tap into the behavior of Greek people.

-The vaccination rates of Greek students on November 2022 were similar to that of the November 2021. Regarding vaccination against Covid-19, similar to the previous years, the majority of students who had not been vaccinated were undergraduates. In was observed here and in previous studies that students who were vaccinated against Covid-19 infection were at higher risk for stress, anxiety and depression in comparison to those who did not receive vaccination against the Covid-19 infection. Early studies on general populations revealed an elevated willingness to undergo vaccination against Covid-19 infection [Eibensteiner]. However, healthcare workers presented a decreased interest on receiving the vaccine, in comparison to general population, and thus, people got reluctant too [cheristanidis snehota]. Of course, there is a variety of factors that contribute to acceptance or not of the vaccination programs (gender, age, education, income, marital status, social media, job, and psychological distress) [cascini, hou, bendau, lazarus]. Moreover, reluctance to get vaccinated against Covid-19 in Greece has been observed mostly in females and less educated respondents [Cheristanidis , choleva]. Parents are particularly positive regarding vaccination, prevention and protection of public health [Cheristanidis]. It appears to be a pattern of prevention and protection during the pandemic in several individuals who were vaccinated against influenza in the past [grochowska].

On that last note, I found your conclusion a bit underwhelming. You note, “The 24-h-communication line that was established for members who seek psychological support or counseling is quite essential based on our findings. The support centers should also consider integrating virtual reality technologies in order to efficiently support these psychopathological symptoms, especially due to the referring population which is consisted of students, who accept rather easily such innovations.” Aside from the awkward English of the last sentence, the conclusion regarding the necessity of support services doesn’t seem to be anything terribly new. Your suggestion about virtual reality technologies arrives rather abruptly here, and without comment or qualification. Has it been shown in the literature that virtual reality interventions enjoy an increased efficacy? If so, perhaps a word or two on how/why.

-Thank you, of course! Universities should also consider integrating virtual reality (VR) technologies in order to efficiently support their students. Several studies reveal that VR is beneficial as a psychological tool for intervention in individuals with mental health problems [hatta and hatta]. Telepsychiatry is a promising and growing way to deliver mental health services [di carlo]. Hatta et al suggest that VR could provide a three-dimensional (3D) ecosystem for people to participate in interactive environments, assist in the training, evaluation, delivery, and supervision of psychotherapy skills [hatta]. Especially in quarantine periods, VR could be a good substitute for public gyms and private group fitness for physical-psychological wellbeing [peng].

We also revised the English.

Overall, I believe your study contributes to the growing literature on the pandemic and mental health. It basically corroborates what others have found. In this regard, I would like to see some more robust explanations for what we find and more explicit recommendations for how to enhance a global response to mental health issues associated with the public health crisis.

-We also performed an extensive English language revision. Thank you so much for your suggestions and significant contribution to improve our manuscript.

Reviewer 2 Report

In this paper, Dimitrios Kavvadas and colleagues assessed stress, anxiety, and depression levels of Aristotle University of Thessaloniki students and their correlations, by the end of the third year of the pandemic.

The present study enquires a very crucial subject in Psychiatry, considering that poor mental health of university students is a growing concern for public health.

There are some issues that I wish the authors would fix to improve the value of the article.

1.   Since the study presented three-year comparative evaluation of mental health and levels of depression, anxiety and stress during the pandemic, it could be interesting to add a brief comment about the interventions expressly tailored to support anxiety and depression in university students during the pandemic. These interventions could have had a big impact on students’ mental health. See this recent review: doi: 10.1016/j.rpsm.2022.04.005

2.     I suggest better describing in the Methods the characteristics of the different waves of the survey collecting data from 2020 and 2021. To make a comparison it is important to know the sample size and other relevant information about the different waves (not only 2022 one), otherwise it is difficult to follow the logic flow of the paper.

3.   Since the questionnaire included also questions about personal perspectives of the pandemic (e.g., “Concern about impending lockdown”), I suggest including in the Introduction a brief comment about student’s perspectives around mental health and Covid-19. See, for example: doi: 10.3390/ijerph20054071

Author Response

In this paper, Dimitrios Kavvadas and colleagues assessed stress, anxiety, and depression levels of Aristotle University of Thessaloniki students and their correlations, by the end of the third year of the pandemic. The present study enquires a very crucial subject in Psychiatry, considering that poor mental health of university students is a growing concern for public health.

-Thank you for your time and effort to review our manuscript.

There are some issues that I wish the authors would fix to improve the value of the article.

  1. Since the study presented three-year comparative evaluation of mental health and levels of depression, anxiety and stress during the pandemic, it could be interesting to add a brief comment about the interventions expressly tailored to support anxiety and depression in university students during the pandemic. These interventions could have had a big impact on students’ mental health. See this recent review: doi: 10.1016/j.rpsm.2022.04.005

-Thank you for your suggestion. We added an extra “intervention” paragraph by the end of discussion section.

  1. I suggest better describing in the Methods the characteristics of the different waves of the survey collecting data from 2020 and 2021. To make a comparison it is important to know the sample size and other relevant information about the different waves (not only 2022 one), otherwise it is difficult to follow the logic flow of the paper.

-Thank you for your suggestion. We added extra information on the sub-paragraph 2.1 study sample and comparison samples.

  1. Since the questionnaire included also questions about personal perspectives of the pandemic (e.g., “Concern about impending lockdown”), I suggest including in the Introduction a brief comment about student’s perspectives around mental health and Covid-19. See, for example: doi: 10.3390/ijerph20054071

-Thank you for your suggestion. We added a brief comment on the third paragraph of the introduction section. 

We also performed an extensive English language revision. Thank you so much for your suggestions and significant contribution to improve our manuscript.

Reviewer 3 Report

The authors present an interesting study on stress, anxiety, and depression observed in students at the University of Thessaloniki in the third year of the pandemic. Scores in the respective scales are compared with those obtained in the previous two years. Further, a number of demographic risk factors are tested. This contribution addressed an important topic and contributes to the growing literature in this field. The sample is large and results are potentially informative. However, I would recommend that the authors address a couple of points in a revision. These pertain to how results are analyzed and results are presented. Further, it should be clarified in some cases what exactly is meant.

Generally, I found it interesting to compare levels of affective disorders across the last three years. What is missing, though, is a pre-pandemic reference level. Hence, it is difficult to say t what extent the level has increased during the pandemic.

The authors used 7-item Likert-type self-report scales to assess the dependent variables. Then, values of these scales are somehow transformed into discrete categories (e.g., "severe"). The setting of cut-off values needs to be validated. Second, a high value on a self-report scale should not be taken as equivalent to a clinical diagnosis. Generally, I would find it more appropriate if the results obtained with continuous self-report scales are reported in terms of M (SD) for the respective demographic subgroups. The authors could still add interpretive notes (e.g., "a value > … would be severe"; at least when somehow validated elsewhere).

Further issues

- In the abstract and at other places: It would be more informative to give effects size of the significant effects rather than p-values.

- l. 31 ff: "their DASS21 score were almost identical to those reported on the previous years of the pandemic, implicating that there are indeed certain risk factors among the demographic characteristics of the participants": The absence of changes in prevalence does not indicate that there are demographic risk factors. This needs to be shown by means of their predictive relations.

- l. 52: "the female gender is heavily weighted during the Covid-19 period the female gender is heavily weighted during the Covid-19 period": I think the authors want to say that females were more likely to display distress and negative emotions.

- l.  73 f: "Based on Aristotle University of Thessaloniki students’ population at a confidence level of 95% and a confidence interval at 3% the sample size needed is almost 1000 students.": I am not sure what exactly was estimated here. Please report the kind of statistical test and the expected effect size in case of a power simulation.

- l.  92: "The Depression-Anxiety-Stress Scale (DASS21) is a widely accepted screening tool": A reference is needed here.

- l.  98: "The Cronbach factor was calculated at 0.951.": I assume Cronbach’s coefficient alpha is meant here. Possibly replace by or add McDonald’s omega as an index of scale reliability.

- l.  108 f: "Previous psychological or psychiatric treatment was reported by 23% of the students.": This sounds relatively high. Was this a convenience sample of students or were some of the participants recruited in ambulances?

- Table 2 gives proportions of students in certain categories of the DASS21: Were the criteria used to split participants into the respective categories somehow validated? Arguably, the prevalence of, e.g., "severe" totally depends on the setting of criteria.

- Table 3: This table is difficult to read because it comprises a lot of numbers. Further, the significance of the chi-squared test does not necessarily imply that there is a "correlation" between the respective variables, as argued by the authors. As there are 3 categories within each scale, this test would be significant also if there is deviation from expected proportions in the middle cells. I would recommend not to split participants into categories but report M (+/-SD) of the continuous scale scores, split by levels of the predictor (in case the predictor is factorial). Technically, a t-test (in case of 2 levels) or one-factorial ANOVA (in case of more than 2 levels), or a correlation coefficient (in case of a continuous predictor; e.g., age) would be more informative.

- l.  135 ff: "The vast majority of students with previous or current psychological or psychiatric treatment and psychoactive drug intake presented a significant prevalence of stress, anxiety and depression (Tables 3 and 4).": Well, this appears relatively obvious... Given that affective disorders were increased during the pandemic, the most likely reason for seeking treatment would be stress, anxiety, and depression.

- Table 4 and 5: How exactly were the odd ratios computed? I assume these are the lowest level vs. a somehow increased level. Again, I do not think odd ratios are a preferred metric when comparing results of continuous scales. Cohen’s d would be more conventional in this case.

- Table 5, Cohabitation: "Alone vs. With 1 person" and "Alone vs. With 2 or more": In case the authors insist on OR statistics, I would rather keep the reference condition identical, i.e., always "…vs. alone". In this case, living "with one" or "with 2 or more" persons would reduce OR below the reference value of 1.00.

- l.  176 ff: "In 2020, anxiety and stress levels presented a normal range (60% and 50% respectively)": I am not sure what exactly these numbers mean (and not either for the following statistics). However, a clinically relevant diagnosis of 60% would be an extremely high prevalence, hence, all but within "a normal range".

- l.  226: "In Europe, students had higher stress, anxiety, and depression compared to the general population.": I do not think this was generally found. However, if students are predominantly female, of a young age, and possibly possess other risk factors, this could potentially contribute to an explanation.  

- l.  232 f: "during the pandemic, European females reported higher stress, anxiety and depression levels compared to European males": This gender effect is relatively well known, already from the time prior to the pandemic. If the authors argue that females suffered more from the pandemic than males, it would need to be shown that there is an interaction effect, i.e., the levels of negative affect increased more highly (relative to pre-pandemic levels) as compared to the increase in males. Further, this gender effect is, most likely, not confined to Europe.

Author Response

The authors present an interesting study on stress, anxiety, and depression observed in students at the University of Thessaloniki in the third year of the pandemic. Scores in the respective scales are compared with those obtained in the previous two years. Further, a number of demographic risk factors are tested. This contribution addressed an important topic and contributes to the growing literature in this field. The sample is large and results are potentially informative.

However, I would recommend that the authors address a couple of points in a revision. These pertain to how results are analyzed and results are presented. Further, it should be clarified in some cases what exactly is meant.

-Thank you for your time and effort to review our manuscript.

Generally, I found it interesting to compare levels of affective disorders across the last three years. What is missing, though, is a pre-pandemic reference level. Hence, it is difficult to say t what extent the level has increased during the pandemic.

-Thank you for your suggestion. We set a pre-pandemic background through studies before the COVID-19 outbreak in Greece. A mild depression prevalence was revealed due to the beginning of the economic crisis, but the results were in most cases conflicting. Moreover, the fact that students, particularly from Greece, are prone to alcohol consumption during their first years of college, could lead to psychological imbalance and negative feelings. On the contrary, another pre-pandemic study assessed the effects of the Mediterranean diet on Greek university students and found reduced levels of depression and stress, due to the consumption of specific local products. We added the above information on a separate paragraph (Discussion).

  • https://doi.org/10.3390/ijerph120504709.
  • http://www.ncbi.nlm.nih.gov/pubmed/24486974.
  • https://doi.org/10.1080/07448481.2021.1996369.
  • https://doi.org/10.1080/07448481.2018.1432625.
  • https://doi.org/10.1002/hpm.2881.
  • https://doi.org/10.1177/0020764019864122.
  • https://doi.org/10.22365/jpsych.2021.025.

The authors used 7-item Likert-type self-report scales to assess the dependent variables. Then, values of these scales are somehow transformed into discrete categories (e.g., "severe"). The setting of cut-off values needs to be validated.

-Thank you for your comment. We added more information in material and methods based on the below paragraphs:

‘‘The DASS21 (Depression, Anxiety, and Stress Scale) was introduced in 1995 by Lovibond and Lovibond. It consists of three self-report scales designed for screening of depression, anxiety, and stress. In 1998, a final version of the DASS that consisted of 21-item (DASS21) was described. Each of the three DASS21 scales contain seven elements, divided into subscales with similar content. The Depression Scale assesses discomfort, despair, life devaluation, self-devaluation, lack of interest/engagement, and inaction. The stress scale assesses autonomic arousal, signs of stress through skeletal muscle movements, stress-induced anxiety, and the subjective experience of anxiety. The stress scale is sensitive to chronic non-specific stimulation. This scale evaluates the difficulty of relaxation, nervous agitation and upset/agitation, the case of an irritable/hyper-reactive characters, and impatience. Scores for depression, anxiety, and stress are calculated by summing the scores for the relevant data. The DASS21 rating scale is used internationally to assess stress, anxiety, and depression levels. It is a recognized and accepted tool by psychologists and psychiatrists with a very good internal consistency. It is a valid Likert-4 scale (0. Not at all, 1. A little, 2. Much, 3. Too much), which calculates the negative emotional states experienced by the participants during the period of time that the survey is available. The Greek version of the DASS-21 scale, based on Greek sources and official translations, was described by Lyrakos et al. This particular version was used in this survey.’’

-The results can be either normal, mild, moderate, severe, or extremely severe. For stress, a normal score is 0–7, 8–9 for mild stress, 10–12 for moderate, 13–16 for severe, and above 17 for extreme severe stress.

Similarly, 0–3 is the normal score for anxiety, 4–5 is the prevalence of the mild anxiety, 6–7 moderate, 8–9 severe and above 10 is the extreme severe anxiety. Finally, a score of 0–4 is normal for depression, 5–6 is mild, 7–10 is moderate, 11–13 is severe, and above 14 is the score for extreme severe depression.

The scores for depression, anxiety, and stress are calculated by summing the scores for the relevant items. We added the information in the methodology section.

STRESS

ANXIETY

DEPRESSION

NORMAL

0-7: 

0-3: 

0-4:

MILD

8-9:

4-5:

5-6:

MODERATE

10-12:

6-7:

7-10:

SEVERE

13-16:

8-9:

11-13:

EXTREMELY SEVERE

17+ : 

10+ :

14+ :

  • Lovibond, S.H.; Lovibond, P.F. Manual for the Depression Anxiety & Stress Scales, 2nd ed.; Sydney Psychology Foundation: Sydney, Australia, 1995.
  • https://doi.org/10.1016/j.jad.2008.01.023.
  • https://doi.org/10.1016/j.psychres.2020.113108.
  • https://doi.org/10.1016/j.psychres.2020.113298.
  • https://doi.org/10.1016/S0924-9338(11)73435-6.
  • https://www.ncbi.nlm.nih.gov/pmc/articles/PMC8998261/

Second, a high value on a self-report scale should not be taken as equivalent to a clinical diagnosis. Generally, I would find it more appropriate if the results obtained with continuous self-report scales are reported in terms of M (SD) for the respective demographic subgroups. The authors could still add interpretive notes (e.g., "a value > … would be severe"; at least when somehow validated elsewhere).

-Thank you for your suggestion. Based on previous studies, we chose to present and analyze our results with the chi-square analysis as was presented on the table that we now removed in the Appendix section. We re-analyze our results and presented the in terms of M (SD) as you kindly suggested (Tables 2 and 3).

Further issues

- In the abstract and at other places: It would be more informative to give effects size of the significant effects rather than p-values.

-Thank you, ORs were added alongside the p-values as one of the common effect size indices (https://www.ncbi.nlm.nih.gov/pmc/articles/PMC3444174/). We re-analyze the tables and present the Cohen’s d.

- l. 31 ff: "their DASS21 score were almost identical to those reported on the previous years of the pandemic, implicating that there are indeed certain risk factors among the demographic characteristics of the participants": The absence of changes in prevalence does not indicate that there are demographic risk factors. This needs to be shown by means of their predictive relations.

-Thank you, we rephrased and means of the predictive relations were calculated. 

- l. 52: "the female gender is heavily weighted during the Covid-19 period the female gender is heavily weighted during the Covid-19 period": I think the authors want to say that females were more likely to display distress and negative emotions.

-Thank you for your kind suggestion. We rephrase according it.  

- l.  73 f: "Based on Aristotle University of Thessaloniki students’ population at a confidence level of 95% and a confidence interval at 3% the sample size needed is almost 1000 students.": I am not sure what exactly was estimated here. Please report the kind of statistical test and the expected effect size in case of a power simulation.

-Thank you, the expected effect size was 1,055 based on a power simulation (It is added on the sample section of material and methods). Based on Aristotle University of Thessaloniki students’ population (92,546 active students by November 2022), at a confidence level of 0.95 and a margin of error 0.03 with the largest standard deviation for a proportion at 0.5, the sample size needed is almost 1,055 students.

- l.  92: "The Depression-Anxiety-Stress Scale (DASS21) is a widely accepted screening tool": A reference is needed here.

-Thank you, reference added.

- l.  98: "The Cronbach factor was calculated at 0.951.": I assume Cronbach’s coefficient alpha is meant here. Possibly replace by or add McDonald’s omega as an index of scale reliability.

-Thank you, McDonald’s omega added.

Mean

SD

Cronbach's α

McDonald's ω

scale

0.923

0.700

0.951

0.951

- l.  108 f: "Previous psychological or psychiatric treatment was reported by 23% of the students.": This sounds relatively high. Was this a convenience sample of students or were some of the participants recruited in ambulances?

-The survey was conducted in the form of an online questionnaire, which was distributed to participants via their official institutional e-mail (“name”@auth.gr). The hosting platform was the official LimeSurvey AUTH (Aristotle University of Thessaloniki) under the supervision of the certified questionnaires authority of Aristotle University). Due to the legislation on personal data protection (GDPR), both the AUTh bioethics committee (Bioethics Committee No. 1.254/20-10-20) as well as the AUTh Data Protection office, granted permission. The LimeSurvey AUTH platform gathered the responses and access was granted only to the head professor of the project (author T.P.) via a personalized link, which created a secure authorized profile. This percentage is high indeed but at some extend is expected, given the fact the previous year (November 2021) the psychological distress of Auth students was immense. Moreover, the Aristotle University of Thessaloniki has established a 24-h-communication line for members who seek counseling and psychological support. Records so far were in alignment with the evidence presented in this study. The cases of the AUTh community members who seek consultation and psychological support has quintupled in a year. Also, the question is about psychological or psychiatric treatment which is a large range and it refers to the previous years (even larger the range) and not current situation.

- Table 2 gives proportions of students in certain categories of the DASS21: Were the criteria used to split participants into the respective categories somehow validated? Arguably, the prevalence of, e.g., "severe" totally depends on the setting of criteria.

-Thank you, as mentioned above the DASS21 rating scale is used internationally to assess stress, anxiety, and depression levels. It is a recognized and accepted tool by psychologists and psychiatrists with a very good internal consistency. It is a valid Likert-4 scale (0. Not at all, 1. A little, 2. Much, 3. Too much), which calculates the negative emotional states experienced by the participants during the period of time that the survey is available. 

-The results can be either normal, mild, moderate, severe, or extremely severe. For stress, a normal score is 0–7, 8–9 for mild stress, 10–12 for moderate, 13–16 for severe, and above 17 for extreme severe stress.

Similarly, 0–3 is the normal score for anxiety, 4–5 is the prevalence of the mild anxiety, 6–7 moderate, 8–9 severe and above 10 is the extreme severe anxiety. Finally, a score of 0–4 is normal for depression, 5–6 is mild, 7–10 is moderate, 11–13 is severe, and above 14 is the score for extreme severe depression.

The scores for depression, anxiety, and stress are calculated by summing the scores for the relevant items.

STRESS

ANXIETY

DEPRESSION

NORMAL

0-7: 

0-3: 

0-4:

MILD

8-9:

4-5:

5-6:

MODERATE

10-12:

6-7:

7-10:

SEVERE

13-16:

8-9:

11-13:

EXTREMELY SEVERE

17+ : 

10+ :

14+ :

  • Lovibond, S.H.; Lovibond, P.F. Manual for the Depression Anxiety & Stress Scales, 2nd ed.; Sydney Psychology Foundation: Sydney, Australia, 1995.
  • https://doi.org/10.1016/j.jad.2008.01.023.
  • https://doi.org/10.1016/j.psychres.2020.113108.
  • https://doi.org/10.1016/j.psychres.2020.113298.
  • https://doi.org/10.1016/S0924-9338(11)73435-6.
  • https://www.ncbi.nlm.nih.gov/pmc/articles/PMC8998261/

- Table 3: This table is difficult to read because it comprises a lot of numbers. Further, the significance of the chi-squared test does not necessarily imply that there is a "correlation" between the respective variables, as argued by the authors. As there are 3 categories within each scale, this test would be significant also if there is deviation from expected proportions in the middle cells. I would recommend not to split participants into categories but report M (+/-SD) of the continuous scale scores, split by levels of the predictor (in case the predictor is factorial). Technically, a t-test (in case of 2 levels) or one-factorial ANOVA (in case of more than 2 levels), or a correlation coefficient (in case of a continuous predictor; e.g., age) would be more informative.

-Thank you for your suggestion. We revised the table based on your comments (Table 3).

- l.  135 ff: "The vast majority of students with previous or current psychological or psychiatric treatment and psychoactive drug intake presented a significant prevalence of stress, anxiety and depression (Tables 3 and 4).": Well, this appears relatively obvious... Given that affective disorders were increased during the pandemic, the most likely reason for seeking treatment would be stress, anxiety, and depression.

-Thank you, we deleted the sentence in order to be more concise.  

- Table 4 and 5: How exactly were the odd ratios computed? I assume these are the lowest level vs. a somehow increased level. Again, I do not think odd ratios are a preferred metric when comparing results of continuous scales. Cohen’s d would be more conventional in this case.

-Thank you, we revised the Table 4 based on your suggestion with the Cohen’s d calculation. Regarding the ORs our results have been categorized based on the scores for depression, anxiety, and stress which were calculated by summing the scores for the relevant items as suggested by previous studies with the DASS21 screening tool.

- Table 5, Cohabitation: "Alone vs. With 1 person" and "Alone vs. With 2 or more": In case the authors insist on OR statistics, I would rather keep the reference condition identical, i.e., always "…vs. alone". In this case, living "with one" or "with 2 or more" persons would reduce OR below the reference value of 1.00.

-Thank you for your suggestion. After the revision of Table 3 we decided that the ORs regarding this specific factor are unnecessary as they provide no further information and, therefore we deleted these lines.

- l.  176 ff: "In 2020, anxiety and stress levels presented a normal range (60% and 50% respectively)": I am not sure what exactly these numbers mean (and not either for the following statistics). However, a clinically relevant diagnosis of 60% would be an extremely high prevalence, hence, all but within "a normal range".

-Thank you, we rephrased the sentence.

- l.  226: "In Europe, students had higher stress, anxiety, and depression compared to the general population.": I do not think this was generally found. However, if students are predominantly female, of a young age, and possibly possess other risk factors, this could potentially contribute to an explanation. 

-Thank you! After re-reviewing several studies, this seems to be the case. Indeed, in most students’ surveys, females were the vast majority. Therefore, we added that to our limitations.

- l.  232 f: "during the pandemic, European females reported higher stress, anxiety and depression levels compared to European males": This gender effect is relatively well known, already from the time prior to the pandemic. If the authors argue that females suffered more from the pandemic than males, it would need to be shown that there is an interaction effect, i.e., the levels of negative affect increased more highly (relative to pre-pandemic levels) as compared to the increase in males. Further, this gender effect is, most likely, not confined to Europe.

-Thank you, we added that to our manuscript.

Thank you so much for your time and effort that you put on our manuscript. Your suggestions were rather valuable. We also performed an extensive English language revision.

Reviewer 4 Report

The manuscript entitled "Stress, anxiety, and depression levels among university students: Three years from the beginning of the pandemic" by Kavvadas et al. aims to assess the level of stress, anxiety and depression in a popolation of greek university students three years after the beginning of the COVID-19 pandemic.

The subject of study is very interesting and the manuscript is suitable to this journal. However, some issues must be solved.

1. Introduction is too short. More information about COVID-19 and the psychological risks associated with pandemics shall be added.

2. Materials and methods are adequate. I would suggest to divide this section in subparagraphs and to insert more information about DASS21 and the various analyses performed.

3. Results are clear and adequately described. Also tables are clear. I would suggest to insert a correlation scheme.

4. Discussion and conclusion are adequate, but there are too little reference. Please add more citations to the article to improve its quality. Futhermore, limitations of the article must be added.

5. Supplementary materials are interesting and adequate.

6. English grammar and spell check by a mothertongue scientist is suggested.

Author Response

The manuscript entitled "Stress, anxiety, and depression levels among university students: Three years from the beginning of the pandemic" by Kavvadas et al. aims to assess the level of stress, anxiety and depression in a popolation of greek university students three years after the beginning of the COVID-19 pandemic.

-Thank you for your time and effort to review our manuscript.

The subject of study is very interesting and the manuscript is suitable to this journal. However, some issues must be solved.

  1. Introduction is too short. More information about COVID-19 and the psychological risks associated with pandemics shall be added.

-Thank you for your suggestion. More information has been added.

  1. Materials and methods are adequate. I would suggest to divide this section in subparagraphs and to insert more information about DASS21 and the various analyses performed.

-Thank you for your suggestion. Sub-divisions and more information have been added.

  1. Results are clear and adequately described. Also tables are clear. I would suggest to insert a correlation scheme.

-Thank you, apart from the graph of the “Moderate to extremely severe stress, anxiety and depression levels of Aristotle University students from November 2020 to November 2022” we also added a graph that depicts Table 2. 

  1. Discussion and conclusion are adequate, but there are too little reference. Please add more citations to the article to improve its quality. Futhermore, limitations of the article must be added.

-Thank you for your suggestion. Limitations were added by the end of the discussion and more references were also added.

  1. Supplementary materials are interesting and adequate.

-Thank you for your comment.

  1. English grammar and spell check by a mothertongue scientist is suggested.

-Thank you, extensive language revisions have been performed.

Round 2

Reviewer 2 Report

The authors replied properly  to reviewer comments and suggestions.

Author Response

-Thank you for your time and effort to review our manuscript.

Reviewer 3 Report

The authors were very responsive and have addressed most points adequately. Only few issued remain that should be addressed.  

- l. 257: Thanks for adding the omega index! However, is this alpha/ omega for the entire set of items? The index should be computed separately for all scales when results are reported also for each scale.

- Line 337: Although readers should be familiar with effect sizes, it possibly facilitates comprehension if conventions (which d or OR would be called small, medium, or large) are briefly given in the text.

- Table 2: Sorry for the confusion! I think Table 2 should comprise proportions (absolute and   relative) frequencies for the respective levels. However, the M(SD) information is better suited for effects of the demographic variables reported in Table 3.

- Table 3: I assume "CI(95%)" refers to the  effect size with its confidence in braces? If yes, possibly indicate "d (95% CI)" in the table and explain in the table note that this indicates Cohen's d (or Hedges g?)

- Table 3 note: "**Extremely Significant (two-tailed P value is < 0.0001)" I assume this should be "p<.001"? Both significance levels could be briefly summarized like the following: "statistical significance levels: *p<.05, **p<.001"?

- Table 4: "Cohen's d" should be moved to the table title. A possible name for the first could could be "demographic variables" or "~ predictors"?

- Thanks for providing information why the proportion of students seeking psychological treatment was this high. Possibly this should be briefly added also in the manuscript. In fact, it might be relevant also to distinguish between "psychological counseling" (i.e., low level treatment) and "therapy" (i.e., more intensive intervention in case of severe impairment).

- l.  477: "there was no significant pre-pandemic background in Greece regarding the stress, anxiety or depression prevalence": Maybe rephrase, e.g., "The prevalence of stress, anxiety and depression were not substantially increased prior to the pandemic"?

- Appendix A: I found this a bit difficult to read. A simple list with left-justified text may be better suited than a table?

Author Response

The authors were very responsive and have addressed most points adequately. Only few issued remain that should be addressed. 

-Thank you for your time and effort to review our manuscript.

- l. 257: Thanks for adding the omega index! However, is this alpha/ omega for the entire set of items? The index should be computed separately for all scales when results are reported also for each scale.

-Thank you, yes, it was for the entire set of DASS21 scales. In the table below you can see the reports of the reliability test based on each item (Stress for S, Anxiety for A and Depression for D) pf the 7-questions three sets. In pdf you will find each report for each set separately. The item-rest correlation is reported above 0.3.

Item Reliability Statistics

If item dropped

Mean

SD

Item-rest correlation

Cronbach's α

McDonald's ω

D7

0.697

1.032

0.656

0.949

0.949

A7

0.776

0.964

0.685

0.949

0.949

A6

0.725

0.949

0.625

0.949

0.950

S7

1.085

1.034

0.672

0.949

0.949

D6

0.872

1.057

0.699

0.948

0.949

D5

0.802

0.981

0.731

0.948

0.948

A5

0.687

0.969

0.749

0.948

0.948

S6

0.759

0.936

0.619

0.950

0.950

D4

1.369

1.073

0.767

0.947

0.948

S5

1.305

1.031

0.738

0.948

0.948

S4

1.296

1.013

0.699

0.948

0.949

D3

1.013

1.096

0.719

0.948

0.948

A4

1.044

1.078

0.656

0.949

0.949

S3

1.240

1.008

0.736

0.948

0.948

A3

0.536

0.865

0.596

0.950

0.950

S2

1.230

1.026

0.691

0.949

0.949

D2

1.177

1.021

0.610

0.950

0.950

A2

0.498

0.823

0.586

0.950

0.950

D1

0.844

0.958

0.754

0.948

0.948

A1

0.471

0.767

0.444

0.952

0.952

S1

0.961

0.908

0.732

0.948

0.948

- Line 337: Although readers should be familiar with effect sizes, it possibly facilitates comprehension if conventions (which d or OR would be called small, medium, or large) are briefly given in the text.

-Thank you, information added on material and methods.

- Table 2: Sorry for the confusion! I think Table 2 should comprise proportions (absolute and   relative) frequencies for the respective levels. However, the M(SD) information is better suited for effects of the demographic variables reported in Table 3.

-Thank you, it is fine! We have kept the absolute and relative frequencies alongside with extra information.

- Table 3: I assume "CI(95%)" refers to the  effect size with its confidence in braces? If yes, possibly indicate "d (95% CI)" in the table and explain in the table note that this indicates Cohen's d (or Hedges g?)

-Sorry for the confusion! CI in Table 3 refers to the confidence interval of the t-test that we applied as you kindly suggested. We added the information at the table title and inside the table. Cohen's d is separately reported on Table 4.

- Table 3 note: "**Extremely Significant (two-tailed P value is < 0.0001)" I assume this should be "p<.001"? Both significance levels could be briefly summarized like the following: "statistical significance levels: *p<.05, **p<.001"?

-Thank you, we corrected it.

- Table 4: "Cohen's d" should be moved to the table title. A possible name for the first could could be "demographic variables" or "~ predictors"?

-Thank you, we corrected both the table title and the column title inside the table.

- Thanks for providing information why the proportion of students seeking psychological treatment was this high. Possibly this should be briefly added also in the manuscript. In fact, it might be relevant also to distinguish between "psychological counseling" (i.e., low level treatment) and "therapy" (i.e., more intensive intervention in case of severe impairment).

-Thank you, we added the information in our discussion (limitations sub-paragraph) and included your suggestion regarding the distinguish for future studies.

- l.  477: "there was no significant pre-pandemic background in Greece regarding the stress, anxiety or depression prevalence": Maybe rephrase, e.g., "The prevalence of stress, anxiety and depression were not substantially increased prior to the pandemic"?

-Thank you!!  We rephrased.

- Appendix A: I found this a bit difficult to read. A simple list with left-justified text may be better suited than a table?

-Thank you once more for your kind suggestions. The presentation of the questionnaire has been revised into a simple list as you suggested (Appendix A)!
